# Clinicopathological and Imaging Features of Breast Papillary Lesions and Their Association with Pathologic Nipple Discharge

**DOI:** 10.3390/diagnostics13050878

**Published:** 2023-02-24

**Authors:** Jeongeum Oh, Ji Yeon Park

**Affiliations:** Department of Radiology, Inje University Ilsan Paik Hospital, Goyang 10380, Republic of Korea

**Keywords:** breast, papillary neoplasm, nipple discharge, mammography, ultrasound

## Abstract

No studies have evaluated whether any clinicopathological or imaging characteristics of breast papillary lesions are associated with pathological nipple discharge (PND). We analyzed 301 surgically confirmed papillary breast lesions diagnosed between January 2012 and June 2022. We evaluated clinical (age of patient, size of lesion, pathologic nipple discharge, palpability, personal/family history of breast cancer or papillary lesion, location, multiplicity, and bilaterality) and imaging characteristics (Breast Imaging Reporting and Data System (BI-RADS), sonographic, and mammographic findings) and compared malignant versus non-malignant lesions and papillary lesions with versus without PND. The malignant group was significantly older than the non-malignant group (*p* < 0.001). Those in the malignant group were more palpable and larger (*p* < 0.001). Family history of cancer and peripheral location in the malignant group were more frequent than in the non-malignant group (*p* = 0.022 and *p* < 0.001). The malignant group showed higher BI-RADS, irregular shape, complex cystic and solid echo pattern, posterior enhancement on ultrasound (US), fatty breasts, visibility, and mass type on mammography (*p* < 0.001, 0.003, 0.009, <0.001, <0.001, <0.001, and 0.01, respectively). On multivariate logistic regression analysis, peripheral location, palpability, and age of ≥50 years were factors significantly associated with malignancy (OR: 4.125, 3.556, and 3.390, respectively; *p* = 0.004, 0.034, and 0.011, respectively). Central location, intraductal nature, hyper/isoechoic pattern, and ductal change were more frequent in the PND group (*p* = 0.003, *p* < 0.001, *p* < 0.001, and *p* < 0.001, respectively). Ductal change was significantly associated with PND on multivariate analysis (OR, 5.083; *p* = 0.029). Our findings will help clinicians examine patients with PND and breast papillary lesions more effectively.

## 1. Introduction

Nipple discharge is a frequently occurring symptom in women, accounting for 2–10% of the symptoms that women present with at breast clinics. Nipple discharge can be classified as a physiologic or a pathologic discharge and is generally considered to be pathologic if it is spontaneous, bloody, clear/serous, unilateral, or arising from a single duct [1]. Common causes of pathologic nipple discharge (PND) are intraductal papilloma, duct ectasia, and malignancy; benign papilloma is the most common etiology (48 to 88%) of PNDs [2,3,4,5,6], followed by duct ectasia (up to 33%) [3], while malignancy accounts for only 5–23% of all PNDs [1,2,3,4,5,7].

Papillary lesions of the breast are varied; these include benign papillomas, atypical papillomas, in situ papillary carcinoma, and invasive papillary carcinomas [8,9]. It is often difficult to distinguish between benign and malignant papillary lesions, because these have been reported to have overlapping clinical symptoms and imaging features [10,11]. Patient’s age, lesion size, multiplicity, and peripheral location of the lesion have been reported to be significant clinical factors, whereas visibility and lesion density on mammography, as well as echo pattern, echogenic halo, orientation, posterior feature, and vascularity on ultrasound, have been reported to be significant radiologic factors, which can differentiate malignant papillary lesions from benign ones [12,13,14,15,16,17,18]. However, previous studies have utilized small sample sizes of less than 200 papillary lesions or have focused on a small number of cases diagnosed as malignancies after being labelled as benign papillomas using core needle biopsies [12,13,14,15,16,17].

To the best of our knowledge, no studies have evaluated whether clinicopathological or imaging characteristics of papillary lesions are associated with pathological nipple discharge (PND).

Consequently, this study aimed to assess whether any clinical or imaging features can assist in the differentiation of breast papillary lesions and to evaluate whether the clinicopathological and imaging findings of breast papillary lesions are associated with PND.

## 2. Materials and Methods

### 2.1. Study Population

Our institutional review board approved this retrospective study and waived the requirement of informed patient consent for the review of medical records and radiologic images.

We searched the pathological database for patients diagnosed with papillary lesions of the breast via surgical excision between January 2012 and June 2022 at our hospital. During this period, 1993 patients underwent surgical excision of breast lesions. Among them, 257 patients with 325 papillary lesions of the breast were identified. We excluded cases that did not have ultrasound (US) and mammography images available, those that had lesions that were not detectable on US or mammography, and those that had incidental papillomas detected on excised specimens for breast cancer. A total of 301 papillary lesions in 238 women were finally included in this study. 

### 2.2. Clinicopathologic Characteristics 

Medical records were reviewed for patient age, clinical symptoms (nipple discharge, palpability), location (central or peripheral), multiplicity, bilaterality, personal or family history (breast cancer, previous papillary lesion), and lesion size, as well as final pathological diagnosis of benign, atypical, or malignant papillary lesions. Pathological nipple discharge was defined as discharge from a single duct or a unilateral; spontaneous; bloody, serous, or clear nipple discharge. Lesions at a distance of ≤2 cm from the nipple were categorized as central and those at >2 cm from the nipple were considered peripheral and the distance was measured on mammography or US. Multiple lesions were defined as ≥2 lesions in the unilateral breast on imaging and pathology. The lesion size was defined as the largest diameter on US.

### 2.3. Imaging Analysis

Mammography was performed using a Lorad Selenia (Hologic, Bedford, MA, USA). US examinations were performed using one of the following high-resolution US systems with an 11–18 MHz linear transducer: HDI 5000 (Philips Healthcare, Bothell, WA, USA), Aixplorer (SuperSonic Imagine, Aix-en-Provence, France), and Aplio I800 (Canon Medical Systems Corporation, Tokyo, Japan). Two radiologists with 2–12 years of experience in breast imaging interpretation retrospectively reviewed the radiologic images by consensus and were blinded to the final pathologic diagnosis. The images were assessed according to the Breast Imaging Reporting and Data System (BI-RADS) lexicon for mammography and US [19]. Breast density, lesion visibility, and lesion type (mass, asymmetry, or calcification) were assessed using mammography. The shape (oval to round, irregular), margin (circumscribed, non-circumscribed), orientation (parallel, nonparallel), echo pattern (hyper-iso, hypo, complex cystic and solid), posterior feature (enhancement, shadowing, absent), calcification (absent, present), and vascularity (absent, present) of the lesion were evaluated on US. Associated ductal change, such as dilatation or continuation with adjacent ducts and the presence of an intraductal feature, were also recorded.

### 2.4. Statistical Analysis

Differences in clinicopathological and imaging variables were compared between the benign/atypical (non-malignant) and malignant groups using Student’s *t*-test, chi-square test, or Fisher’s exact test. Differences between the PND and non-PND groups were also evaluated. Multiple linear regression analysis was used to evaluate significant factors associated with malignancy and PND. Statistical analyses were performed using SPSS 25.0 (IBM, Armonk, NY, USA) and, when the *p*-value was less than 0.05, it was considered to be statistically significant.

## 3. Results

A total of 301 papillary lesions in 238 women (mean age: 49.31 ± 12.22 years, range: 12–88 years) were included in this study. Twenty-one women had bilateral papillary lesions, 13 had multiple lesions present simultaneously, and 22 had relapsed papillary lesions within the study period.

There were 192 (63.8%) benign papillary lesions, 68 (22.6%) atypical lesions, and 41 (13.6%) malignant lesions. Furthermore, 55 (18.3%) papillary lesions were associated with PND and 246 (81.7%) were not.

### 3.1. Comparison of Clinical and Imaging Characteristics between Non-Malignant and Malignant Papillary Lesions

Among the clinical factors, age, palpability, lesion size, family history of cancer, and location were significantly different between the non-malignant and malignant groups (Table 1). The mean ages of the benign, atypical, and malignant papillary lesion groups were 46.89 ± 11.18, 49.96 ± 11.49, and 59.56 ± 12.88 years, respectively; additionally, patients in the malignant group were significantly older than those in the non-malignant group (*p* < 0.001). In the benign, atypical, and malignant lesion groups, 32.3% (62/192), 35.3% (24/68), and 68.3% (29/41), respectively, were ≥50 years of age (*p* < 0.001). Lesion size on US was available for 295 cases: the mean sizes of benign, atypical, and malignant papillary lesions were 0.96 ± 0.65 cm, 0.90 ± 0.43 cm, and 1.76 ± 1.36 cm, respectively. Moreover, lesions in the malignant group were significantly larger than those in the non-malignant group (*p* < 0.001). The proportion of lesions sized ≥1 cm were 32.1% (61/190), 46.9% (30/64), and 65.9% (27/41) in the benign, atypical, and malignant groups, respectively, (*p* < 0.001). The proportion of lesions that were palpable were 9.4% (18/192), 10.3% (7/68), and 51.2% (21/42) in the benign, atypical, and malignant lesion groups, respectively. Malignant papillary lesions were more frequently palpable than non-malignant lesions (*p* < 0.001). Family history of cancer (12.2%, 5/41) and peripheral location (46.3%, 19/41) in the malignant group were more frequent than in the non-malignant group (3.8%, 21.5%, respectively) (*p* = 0.022 and *p* < 0.001, respectively) (Figure 1). Factors such as pathological nipple discharge, history of breast cancer, previous papillary lesions, multiplicity, and bilaterality were not statistically significant.

Among the imaging characteristics, BI-RADS category, shape, echo pattern, posterior feature on US, breast density, visibility, and lesion type on mammography were significantly different between the non-malignant and malignant groups (Table 2). The malignant group showed a higher BI-RADS than the non-malignant group. Moreover, 51.2% (21/41) of malignancies were categorized as 4B, 4C, and 5 and 88.1% (229/260) of the lesions in the non-malignant group were scored as 3 and 4A (*p* < 0.001). Additionally, 53.7% (22/41) of malignant lesions had an irregular shape and 69.7% (177/254) of non-malignant lesions had an oval to round shape (*p* = 0.003). A complex cystic and solid echo pattern was found in 24.4% (10/41) of malignant lesions and hyper/iso echotexture was found in 19.2% (49/254) of non-malignant lesions; the proportion of hypoechoic patterns was similar between the two groups (*p* = 0.009). Posterior enhancement was more frequent in the malignant group (48.8%, 20/41) than in the non-malignant group (*p* < 0.001). 

Of the 227 cases with available mammographic data, fatty breasts were more frequent in cases of malignancies (56.8%, 21/37) and dense breasts were common in cases of non-malignancies (86.8%, 165/190) (*p* < 0.001). The malignant lesions were significantly more visible on mammography (86.5%, 32/37) (*p* < 0.001); 75.0% of malignancies were masses and 57.6% (38/66) of non-malignancies were presented as asymmetry or calcification only (*p* = 0.01).

Factors such as the presence of intraductal features, margins, orientation, calcification, vascularity, and ductal change on US were not statistically significant.

In the multivariate logistic regression analysis, peripheral location, palpability, and patient age of ≥50 years were significant factors associated with malignant papillary lesions (OR: 4.125, 3.556, and 3.390, respectively; *p* = 0.004, 0.034, and 0.011, respectively) (Table 3).

### 3.2. Comparison of Clinicopathological and Imaging Characteristics of Papillary Lesions between PND and Non-PND Groups

Table 4 and Table 5 show the comparison of papillary lesion with or without PND. The location, echo pattern, presence of intraductal features, and ductal changes were statistically significant between the two groups. Central location in the PND group was more frequent than in the non-PND group (90.9% vs. 71.5%, respectively, *p* = 0.003). In the PND group, 34.5% (19/55) of lesions were hyper/isoechoic, 60% (33/55) were hypoechoic, and 5.5% (3/55) were complex echoic. In the non-PND group, 14.2% (34/240) of lesions were hyper/isoechoic, 73.3% (176/240) were hypoechoic, and 12.5% (30/240) were complex echoic (*p* < 0.001). Ductal change and intraductal feature were more common in the PND group than in the non-PND group (*p* < 0.001) (Figure 2). Other factors, including the pathological results, showed no significant association with PND.

Ductal change was the only significant factor associated with PND in the multivariate logistic regression analysis (OR: 5.083, *p* = 0.029) (Table 6).

## 4. Discussion

Benign papilloma is the most common cause of PND and has been reported to account for up to 88% of PND cases [2,3,4,5,6]. However, (i) the association between the pathology of papillary breast lesions and PND and (ii) the variable clinical or imaging factors affecting PND are unknown. There is a lack of consensus with respect to the management of papillary breast lesions because of differences in populations sampled and the methods employed by the different studies [15,16,17,20]. To the best of our knowledge, our study is the largest series investigating papillary lesions along with their surgical results and the first study focusing on the association between papillary lesions and PND.

Our study revealed that old age (≥50 years), palpability of the lesion, family history of breast cancer, large lesion size (≥1 cm), and peripheral location of the lesion were clinically associated with malignant papillary lesions. These results are consistent with those of previous studies that reported that peripheral location, lesion size, and old age are correlated with malignancy [13,17,18,21,22]. Palpability was also reported to be a significant factor associated with malignant papillary lesions (50%, 5/10) [11,23]. However, our results regarding family history of breast cancer were discordant with those of previous studies. A study reported no significant differences in family history between benign and atypical/malignant papillary lesions [24]. One study showed a trend toward a higher upgrade rate to a malignancy in patients with a family history of breast cancer, without statistical significance (*p* = 0.09) [25]. In another study evaluating benign papillomas diagnosed by core biopsy, family history was not associated with an upgraded diagnosis to malignancy (n = 14) [21]. These studies had a limited number of cases of malignancies (up to 21 cases); therefore, a study with a larger number of papillary lesions is necessary in the future.

In the present study, high BI-RADS, irregular shape, complex echogenicity, and posterior enhancement were the sonographic characteristics associated with malignancy. These results are supported by previous studies, and high BI-RADS scores were one of the significant factors in predicting malignant progression [17,18,23]. A complex echo pattern was reported to be significantly frequent in malignant lesions [13]. Several studies suggested that margins, echo pattern, and posterior features were significant sonographic features that assisted in differentiating papillary lesions [14,15,26]. 

Our study showed that fatty breasts, visibility, and mass type were mammographic features significantly associated with malignancy. Old patients tend to have fatty breast density on mammography, and background fatty density and the large size of malignant papillary lesions may partly explain the greater visibility of the lesions on mammography. In a previous study, fatty breast density was reported to be a predictor of upgrade to malignancy [27]. Visibility on mammography as a significant factor in predicting malignant papillary lesions was already proven by a previous study [13]. Mass on mammography has been reported to be a risk factor for upgrade to malignancy in benign papillomas, diagnosed using core biopsy [11,23].

The factors significantly associated with PND for papillary lesions in our study were central location of the lesion, hyperechoic/isoechoic pattern, intraductal lesion, and ductal changes on ultrasound. Because common causes of PND are intraductal papilloma or duct ectasia and solitary papillomas usually present as subareolar duct dilatation with internal solid echoes [28,29], these results are sufficiently predictable. Therefore, when a physician encounters a patient with PND, an intraductal echogenic lesion with central location or lesion with ductal changes on US may be predicted.

This study showed that the surgical pathology of papillary lesions was not associated with PND. Among the 55 papillary lesions with PND, 85.4% of them were benign/atypical lesions and malignancy was only found in 14.5%. According to published studies, the predictive value of nipple discharge for malignant papillary lesions remains unclear. A study including 51 papillary lesions with surgical results reported that 32.4% of benign/atypical papillomas and 0% of malignant papillary lesions presented with a pathological nipple discharge [14]. Wang et al. demonstrated that bloody nipple discharge is a significant indicator of papilloma with high-risk or malignant lesions (*p* = 0.009) [24]. Ahn et al. reported that bloody nipple discharge was significantly associated with upgraded malignancy in benign papillomas diagnosed by core biopsy [23]. Rizzo et al. reported that 16.2% of papillary lesions upgraded to a diagnosis of malignancy had bloody nipple discharge, but the difference was not statistically significant [30]. However, these studies had only a small number of malignant cases or included only benign papillomas diagnosed by core biopsy.

Our study has several limitations. First, this was a retrospective study conducted at a single institution, and the number of papillary breast lesions with pathological nipple discharge was small. Therefore, multicenter studies with larger sample sizes are necessary. Second, we considered multiple or recurrent lesions of each patient to be separate cases; this could have affected the results. Third, seven lesions of patients who had other concurrent malignancy in the ipsilateral breast were not excluded because the patients did not have PND. However, the strength of our study is that all lesions were surgically excised and histopathologically confirmed. Additionally, this is the first study to focus on the association between pathological nipple discharge and the clinicopathological or imaging features of breast papillary lesions.

## 5. Conclusions

Among breast papillary lesions, old age, palpability of lesion, large lesion size, peripheral location of the lesion, and a family history of breast cancer were found to be clinically associated with malignancy. A high BI-RADS category, irregular shape, complex echo, posterior enhancement on US, fatty breasts, mammographic visibility, and type of mass on mammography were also imaging characteristics associated with malignancy. Hyper/iso echo pattern, central location, presence of intraductal feature, and ductal change were significantly associated with PND. Consequently, we propose that breast physicians and radiologists should pay particular attention to the clinical history, physical examination, and imaging findings when they encounter patients with PND and breast papillary lesions.

## Figures and Tables

**Figure 1 diagnostics-13-00878-f001:**
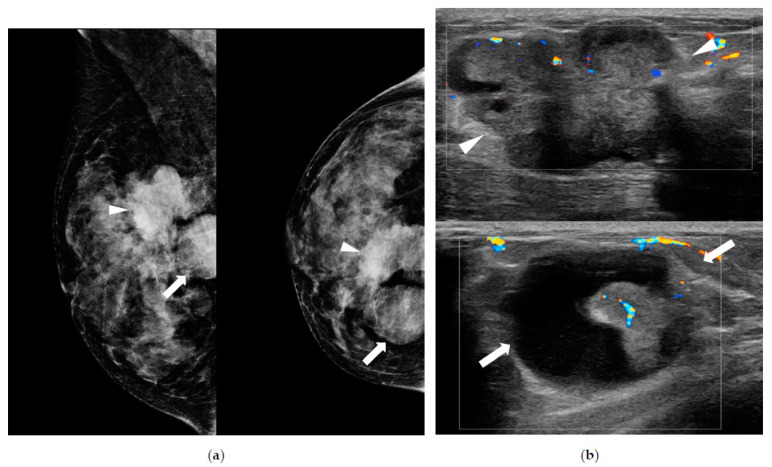
A 46-year-old woman presented with palpable masses in the breast accompanied by bloody nipple discharge. (**a**) Mammography shows a hyperdense mass with spiculated margins and an irregular shape in the upper central portion (arrowheads) and a circumscribed round hyperdense mass in the upper inner portion of right breast (arrows). (**b**) Ultrasound shows a 3.9 cm heterogeneous echoic mass (arrowheads) with irregular shape and internal vascularity in the right breast, in the 1 o’clock position, 5 cm from the nipple. Another 2.7 cm complex cystic and solid mass (arrows) is noted in the right breast in the 2 o’clock position, 6 cm from the nipple. Ductal carcinoma in situ with focal invasion in the background of papilloma was confirmed by surgery. The patient had a family history of breast cancer (mother).

**Figure 2 diagnostics-13-00878-f002:**
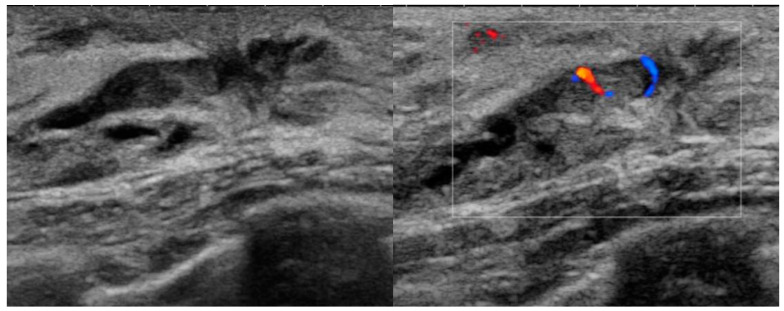
A 38-year-old woman presented with bloody nipple discharge. Ultrasound shows a 1.0 cm isoechoic intraductal mass with internal blood flow in the left breast subareolar area. Benign intraductal papilloma was confirmed on pathology after surgical excision.

**Table 1 diagnostics-13-00878-t001:** Clinical characteristics of papillary lesions according to pathological examination.

	Total	Benign Group	Atypical Group	Malignant Group	*p*-Value *
Age (years)	49.31 ± 12.22 (12–88)	46.89 ± 11.18 (12–88)	49.96 ± 11.49 (28–74)	59.56 ± 12.88 (35–85)	<0.001
Age group (n) Age < 50 years Age ≥ 50 years	301 187 (62.1%) 114 (37.9%)	192 130 (67.7%) 62 (32.3%)	68 44 (64.7%) 24 (35.3%)	41 13 (31.7%) 28 (68.3%)	<0.001
Lesion size (cm)	1.06 ± 0.80 (0.3–6.8)	0.96 ± 0.65 (0.3–4.5)	0.90 ± 0.43 (0.3–2.0)	1.76 ± 1.36 (0.3–6.8)	<0.001
Lesion size group (n) <1 cm ≥1 cm	295 177 (58.80%) 118 (39.20%)	190 129 (67.9%) 61 (32.1%)	64 34 (53.1%) 30 (46.9%)	41 14 (34.1%) 27(65.9%)	0.001
Palpability (n) (−) (+)	301 255 (84.7%) 46 (15.30%)	192 174 (90.6%) 18 (9.4%)	68 61 (89.7%) 7 (10.3%)	41 20 (48.8%) 21 (51.2%)	<0.001
PND (n) (−) (+)	301 246 (81.7%) 55 (18.3%)	192 152 (79.2%) 40 (20.8%)	68 61 (89.7%) 7 (10.3%)	41 33 (80.5%) 8 (19.5%)	0.825
Multiplicity (n) Single Multiple	301 272 (90.40%) 29 (9.60%)	192 174 (90.6%) 18 (9.4%)	68 63 (92.6%) 5 (7.4%)	41 35 (85.4%) 6 (14.6%)	0.243
Bilaterality (n) Unilateral Bilateral	301 258 (85.70%) 43 (14.30%)	192 164 (85.4%) 28 (14.6%)	68 59 (86.8%) 9 (13.2%)	41 35 (85.4%) 6 (14.6%)	0.945
Location (n) Central Peripheral	301 226 (75.1%) 75 (24.9%)	192 157 (81.8%) 35 (18.2%)	68 47 (69.1%) 21 (30.9%)	41 22 (53.7%) 19 (46.3%)	<0.001
Previous history of cancer (n) (−) (+)	301 292 (97.0%) 9 (3.0%)	192 188 (97.9%) 4 (2.1%)	68 64 (94.1%) 4 (5.9%)	41 40 (97.6%) 1 (2.4%)	>0.999
Family history of cancer (n) (−) (+)	301 286 (95.0%) 15 (5.0%)	192 183 (95.3%) 9 (4.7%)	68 67(98.5%) 1 (1.5%)	41 36(87.8%) 5 (12.2%)	0.022
Previous papillary lesion (n) (−) (+)	301 267 (88.7%) 34 (11.30%)	192 167 (87.0%) 25 (13.0%)	68 62 (91.2%) 6 (8.8%)	41 38 (92.7%) 3 (7.3%)	0.595

* Malignant group vs. benign and atypical groups.

**Table 2 diagnostics-13-00878-t002:** Imaging characteristics of papillary lesions according to final pathology.

	Total	Benign Group	Atypical Group	Malignant Group	*p*-Value *
BI-RADS (n) Category 3 Category 4A Category 4B Category 4C Category 5	301 7 (2.30%) 242 (80.4%) 31 (10.3%) 12 (4.0%) 9 (3.0%)	192 4 (2.1%) 168 (87.5%) 18 (9.4%) 2 (1.0%) 0 (0%)	68 3 (4.4%) 54 (79.4%) 7 (10.3%) 2 (2.9%) 2 (2.9%)	41 0 (0%) 20 (48.8%) 6 (14.6%) 8 (19.5%) 7 (17.1%)	<0.001
Shape on US (n) Oval to round Irregular	295 196 (65.10%) 99 (32.90%)	190 133 (70.0%) 57 (30.0%)	64 44 (68.8%) 20 (31.3%)	41 19 (46.3%) 22 (53.7%)	0.003
Margin on US (n) Circumscribed Non-circumscribed	295 99 (32.90%) 196 (65.10%)	190 69 (43.1%) 121 (63.7%)	64 21 (32.8%) 43 (67.2%)	41 9 (22.0%) 32 (78.0%)	0.09
Echo on US (n) Hyper/iso Hypo Complex	295 53 (17.60%) 209 (69.40%) 33 (11.0%)	190 39 (20.5%)$ 136 (71.6%) 15 (7.9%)	64 10 (15.6%) 46 (71.9%) 8 (12.5%)	41 4 (9.8%) 27 (65.9%) 10 (24.4%)	0.009
Orientation on US (n) Parallel Non-parallel	295 268 (89.00%) 27 (9.0%)	190 175 (92.1%) 15 (7.9%)	64 55 (85.9%) 9 (14.1%)	41 38 (92.7%) 3 (7.3%)	>0.999
Posterior feature on US (n) Enhancement Shadowing No	295 71 (23.60%) 4 (1.30%) 220 (73.10%)	190 38 (20.0%) 2 (1.1%) 150 (78.9%)	64 13 (20.3%) 1 (1.6%) 50 (78.1%)	41 20 (48.8%) 1 (2.4%) 20 (48.8%)	<0.001
Calcification on US (n) (−) (+)	295 281 (93.40%) 14 (4.70%)	190 183 (96.3%) 7 (3.7%)	64 59 (92.2%) 5 (7.8%)	41 39 (95.1%) 2 (4.9%)	>0.999
Vascularity on US (n) (−) (+)	286 79 (26.20%) 207 (68.80%)	186 58 (31.2%) 128 (68.8%)	61 14 (23.0%) 47 (77.0%)	39 7 (17.9%) 32 (82.1%)	0.27
Intraductal feature on US (n) (−) (+)	295 186 (61.80%) 109 (36.20%)	190 111 ((58.4%) 79 (41.6%)	64 45 (70.3%) 19 (29.7%)	41 30 (73.2%) 11 (26.8%)	0.148
Ductal change on US (n) (−) (+)	295 180 (59.80%) 115 (38.20%)	190 107 (56.3%) 83 (43.7%)	64 45 (70.3%) 19 (29.7%)	41 28 (68.3%) 13 (31.7%)	0.303
Density on MG (n) Fatty Dense	227 46 (15.30%) 181 (60.10%)	136 16 (11.8%) 120 (88.2%)	54 9 (16.7%) 45 (83.3%)	37 21 (56.8%) 16 (43.2%)	<0.001
Visibility on MG (n) (−) (+)	227 129 (56.8%) 98 (43.2%)	136 95 (69.9%) 41 (30.1%)	54 29 (53.7%) 25 (46.3%)	37 5 (13.5%) 32 (86.5%)	<0.001
Type on MG Mass Asymmetrical Calcification only	98 52 (53.0%) 38 (38.8%) 8 (8.2%)	41 15 (36.6%) 23 (56.1%) 3 (7.3%)	25 13 (52.0%) 8 (32.0%) 4 (16.0%)	32 24 (75.0%) 7 (21.9%) 1 (3.1%)	0.01

* Malignant group vs. benign and atypical lesion groups. BI-RADS, Breast Imaging Reporting and Data System; MG, mammography; US, ultrasound.

**Table 3 diagnostics-13-00878-t003:** Multivariate logistic regression analysis to determine the risk factors associated with malignant papillary breast lesions.

Variable	Odds Ratio (95% CI)	*p*-Value
Location Peripheral	4.125 (1.582–10.753)	0.004
Palpability	3.556 (1.103–11.470)	0.034
Age group ≥50 years	3.390 (1.327–8.661)	0.011

CI: confidence interval.

**Table 4 diagnostics-13-00878-t004:** Clinical characteristics of papillary lesions, classified according to pathologic nipple discharge.

	PND Group (n = 55)	Non-PND Group (n = 246)	*p*-Value
Age (years)	47.42 ± 12.88 (range: 12–74 years)	49.73 ± 12.05 (range: 19–88 years)	0.205
Age group <50 years ≥50 years	36 (65.5%) 19 (34.5%)	151 (61.4%) 95 (38.6%)	0.574
Lesion size	1.20 ± 0.88 (range: 0.4–5.0)	1.03 ± 0.78 (0.3–6.8)	0.163
Lesion size group (n) <1 cm ≥1 cm	55 29 (52.7%) 26 (47.3%)	240 148 (61.7%) 92 (38.3%)	0.222
Pathology Benign Atypical Malignancy	40 (72.7%) 7 (12.7%) 8 (14.5%)	152 (61.8%) 61 (24.8%) 33 (13.4%)	0.151
Location Central Peripheral	50 (90.9%) 5 (9.1%)	176 (71.5%) 70 (28.5%)	0.003
Multiplicity Single Multiple	49 (89.1%) 6 (10.9%)	223 (90.7%) 23 (9.3%)	0.723
Bilaterality Unilateral Bilateral	48 (87.3%) 7 (12.7%)	36 (14.6%) 210 (85.4%)	0.715
History of previous cancer (−) (+)	54 (98.2%) 1 (1.8%)	238 (96.7%) 8 (3.3%)	>0.999
Family history of cancer (−) (+)	50 (90.9%) 5 (9.1%)	236 (95.9%) 10 (4.1%)	0.121
Previous papillary lesion (−) (+)	53 (96.4%) 2 (3.6%)	214 (87.0%) 32 (13.0%)	0.057

PND, pathologic nipple discharge.

**Table 5 diagnostics-13-00878-t005:** Imaging characteristics of papillary lesions, classified according to pathologic nipple discharge.

	PND Group (n = 55)	Non-PND Group (n = 246)	*p*-Value
BI-RADS Category 3 Category 4A Category 4B Category 4C Category 5	1 (1.8%) 41 (74.5%) 9 (16.4%) 1 (1.8%) 3 (5.5%)	6 (2.4%) 201 (81.7%) 22 (8.9%) 11 (4.5%) 6 (2.4%)	0.295
Shape on US (n) Oval to round Irregular	55 38 (69.1%) 17 (30.9%)	240 158 (65.8%) 82 (34.2%)	0.644
Margin on US (n) Circumscribed Non-circumscribed	55 23 (41.8%) 32 (58.2%)	240 76 (31.7%) 164 (68.3%)	0.150
Echo on US (n) Hyper/iso Hypo Complex	55 19 (34.5%) 33 (60.0%) 3 (5.5%)	240 34 (14.2%) 176 (73.3%) 30 (12.5%)	0.001
Orientation on US (n) Parallel Non-parallel	55 51 (92.7%) 4 (7.3%)	240 217 (90.4%) 23 (9.6%)	0.796
Posterior feature on US (n) Enhancement Shadowing No	55 11 (20.0%) 0 44 (80.0%)	240 60 (25.0%) 4 (1.7%) 176 (73.3%)	0.439
Calcification on US (n) (−) (+)	55 51 (92.7%) 4 (7.3%)	240 230 (95.8%) 10 (4.2%)	0.304
Vascularity on US (n) (−) (+)	54 10 (18.5%) 44 (81.5%)	232 69 (29.7%) 163 (70.3%)	0.209
Ductal change on US (n) (−) (+)	55 13 (23.6%) 42 (76.4%)	240 167 (69.6%) 73 (30.4%)	<0.001
Intraductal feature on US (n) (−) (+)	55 15 (27.3%) 40 (72.7%)	240 171 (71.3%) 69 (28.8%)	<0.001
Density on MG (n) Fatty Dense	47 11 (23.4%) 36 (76.6%)	180 35 (19.4%) 145 (80.6%)	0.548
Visibility on MG (n) (−) (+)	47 25 (53.2%) 22 (46.8%)	180 104 (57.8%) 76 (42.2%)	0.572
Type on MG (n) Mass Asymmetry Calcification only	22 9 (40.9%) 12 (54.5%) 1 (4.5%)	76 43 (56.6%) 26 (34.2%) 7 (9.2%)	0.216

PND, pathologic nipple discharge; BI-RADS, Breast Imaging Reporting and Data System; MG, mammography; US, ultrasound.

**Table 6 diagnostics-13-00878-t006:** Multivariate logistic regression analysis to determine the risk factors associated with pathologic nipple discharge in papillary breast lesions.

Variable	Odds Ratio (95% CI)	*p*-Value
Ductal change	5.083 (1.180–21.894)	0.029

CI: confidence interval.

## Data Availability

Data generated in this study are available upon request from the corresponding author.

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
