# Peer review of "Clinicopathological and Imaging Features of Breast Papillary Lesions and Their Association with Pathologic Nipple Discharge"

_diagnostics, 2023, doi:10.3390/diagnostics13050878_

Round 1
Reviewer 1 Report
Benign and malignant lesions of the breast are becoming more and more prevalent. New means to diagnose and segregate such lesions are necessary, making the current article valuable for the medical society. Combining clinical and imaging features to distinguish benign from malignant is entirely original. I congratulate the authors for their work. I suggest reassessing the entire material for the English language.
I found a minor issue in Table 1. In the line describing women above 50 years, the sum is not 114 (or the numbers must be corrected).
Author Response
Reviewer 1
English language and style
(x) English language and style are fine/minor spell check required.
Comments and Suggestions for Authors
Benign and malignant lesions of the breast are becoming more and more prevalent. New means to diagnose and segregate such lesions are necessary, making the current article valuable for the medical society. Combining clinical and imaging features to distinguish benign from malignant is entirely original. I congratulate the authors for their work. I suggest reassessing the entire material for the English language.
I found a minor issue in Table 1. In the line describing women above 50 years, the sum is not 114 (or the numbers must be corrected).
Response: We would like to thank the reviewer. We checked your comment and revised it (Table 1). The number of Age ≥ 50 years in Malignant group was corrected to 28.
And we got it corrected by a professional English language editing company (Editage) according to your advice. We hope that the language quality level has been significantly improved in the revised manuscript.

Reviewer 2 Report
According to the authors, “no studies have evaluated whether clinicopathological or imaging characteristics of papillary lesions are associated with pathological nipple discharge (PND),” so this manuscript is of novelty. The manuscript is generally well written. It is suitable for publication in Diagnostics.
Author Response
Authors’ response to the reviewers’ comments
Reviewer 2
English language and style
(x) English language and style are fine/minor spell check required.
Comments and Suggestions for Authors
According to the authors, “no studies have evaluated whether clinicopathological or imaging characteristics of papillary lesions are associated with pathological nipple discharge (PND),” so this manuscript is of novelty. The manuscript is generally well written. It is suitable for publication in Diagnostics.
Response: We would like to thank the reviewer. And we got it corrected by a professional English language editing company (Editage) according to your advice. We hope that the language quality level has been significantly improved in the revised manuscript.
